# The efficacy of oxidized regenerated cellulose (SurgiGuard®) in breast cancer patients who undergo total mastectomy with node surgery: A prospective randomized study in 94 patients

Kug Hyun Nam [1], Joon-Hyop Lee[1], Yoo Seung Chung[1], Yong Soon Chun[1,2], Heung Kyu Park[1,2], Yun Yeong Kim [1,2]*

1 Department of General Surgery, Gachon University College of Medicine, Incheon, Republic of Korea,
2 Department of General Surgery, Breast Cancer Center, Gachon University Gil Medical Center, Incheon, Republic of Korea

* crysblue511@gmail.com

## Abstract

### Background

Seromas frequently develop in patients who undergo total mastectomy with node surgery. We aimed to prospectively explore whether use of oxidized regenerated cellulose (ORC, SurgiGuard®) affects seroma formation after total mastectomy with node surgery (sentinel lymph node biopsy (SLNB) or axillary lymph node dissection (ALND)).

### Materials and methods

Ninety four breast cancer patients were enrolled in the study who underwent total mastectomy with ALND or SLNB. The patients were randomized into two groups, one treated with ORC plus closed suction drainage and the other with closed suction drainage alone.

### Results

Mean drainage volume was slightly lower in the ORC group on postoperative day 1 (123 ± 54 vs 143 ± 104 ml), but was slightly higher at all other time points; however, these differences were not significant. Mean total drainage volume in patients treated with ORC plus drainage did not differ from that of patients treated with drainage alone (1134 ± 507 ml vs 1033 ± 643 ml, $P = 0.486$).

### Conclusions

Use of ORC (SurgiGuard®) did not significantly alter the risk of seroma formation.

**Data Availability Statement:** The study's minimal data are available at public repository (http://doi.org/10.5281/zenodo.4784906).

**Funding:** This work was supported by Gachon University research fund of 2017 (GCU-2017-5258). The funders had no role in study design, data collection and analysis, decision to publish, or preparation of the manuscript.

**Competing interests:** The authors have declared that no competing interests exist.

## Introduction

The most common complication following mastectomy is seroma formation, with an incidence ranging from 15% to 85% [1,2]. Seroma is a serous fluid collection which may develop in the space between the chest wall and skin flaps. Complications from breast surgical procedures could be costly and may delay subsequent adjuvant therapies. Seroma may be the commonest early sequel, albeit minor consequence, prolongs recovery, length of hospital stay and over stretch budget. Postoperative seroma following mastectomy raises possibilities of wound infection, impaired wound healing, cellulitis and skin necrosis. Patients with wound seroma routinely return to outpatient clinics for aspirations and drainage, subsequent complications such as infection may follow.

The dissection of small lymphatic and blood vessels during breast and axillary surgery can result in an environment highly suitable for fluid collection, although the pathophysiologic processes underlying seroma formation have not been fully determined. The pathogenesis of seroma has not been fully demonstrated. Theories of postoperative seroma is elucidated by acute inflammatory exudates in response to surgical trauma and acute phase of wound healing [3,4]. Several surgical techniques have been used to reduce seroma formation, including the use of ultrasonic scissors, the physical closure of dead spaces, suction drainage, and the placement of external compression dressings [1,5–7]. To date, however, no single method has been found to consistently and reliably prevent seroma formation.

Oxidized regenerated cellulose (ORC) is a topical hemostatic agent that has been in use for several decades [8,9]. ORC acts hemostatically by absorbing blood, by surface interaction with platelets and proteins, and by activating the coagulation cascade. A novel ORC system, Surgi-Guard® (Samyang Biopharmaceuticals Corp., Seoul, Korea), has been approved as a hemostatic agent by the Korean Food and Drug Administration (FDA).

This study hypothesized that the hemostatic properties of this ORC would accelerate wound healing following breast and axillary surgery, including the removal of breast and axillary lymph nodes. ORC could reduce the accumulation of fluid resulting from transection of multiple small blood vessels and lymphatics, thereby reducing the duration and amount of serosanguinous drainage. The present study therefore compared the effects of SurgiGuard® plus closed suction drainage with those of suction drainage alone on seroma formation in patients undergoing total mastectomy and node surgery. High output and/or prolonged drainage tube use was regarded as an indication of increased risk of seroma formation.

## Methods

### Trial design and any changes after trial commencement

This study was a single blinded, prospective randomized controlled trial conducted at a single specialized breast cancer center. Efficacy of ORC is established by demonstrating its superiority to a controlled arm.

### Participants, eligibility criteria, and settings

Patients scheduled to undergo total mastectomy and node surgery (axillary lymph node dissection or sentinel lymph node biopsy) between June 2019 and July 2020 were enrolled in the study. This study was registered at Clinical Research Information Service (CRiS), Republic of Korea (KCT0005637). Study should have been registered before enrollment of participants, but it has been delayed due to systemic errors. Study registration teamwork was so independent upon clinical study process that the protocol flow itself was not interrupted by this late registration. The authors confirm that all ongoing and related trials for this drug/intervention

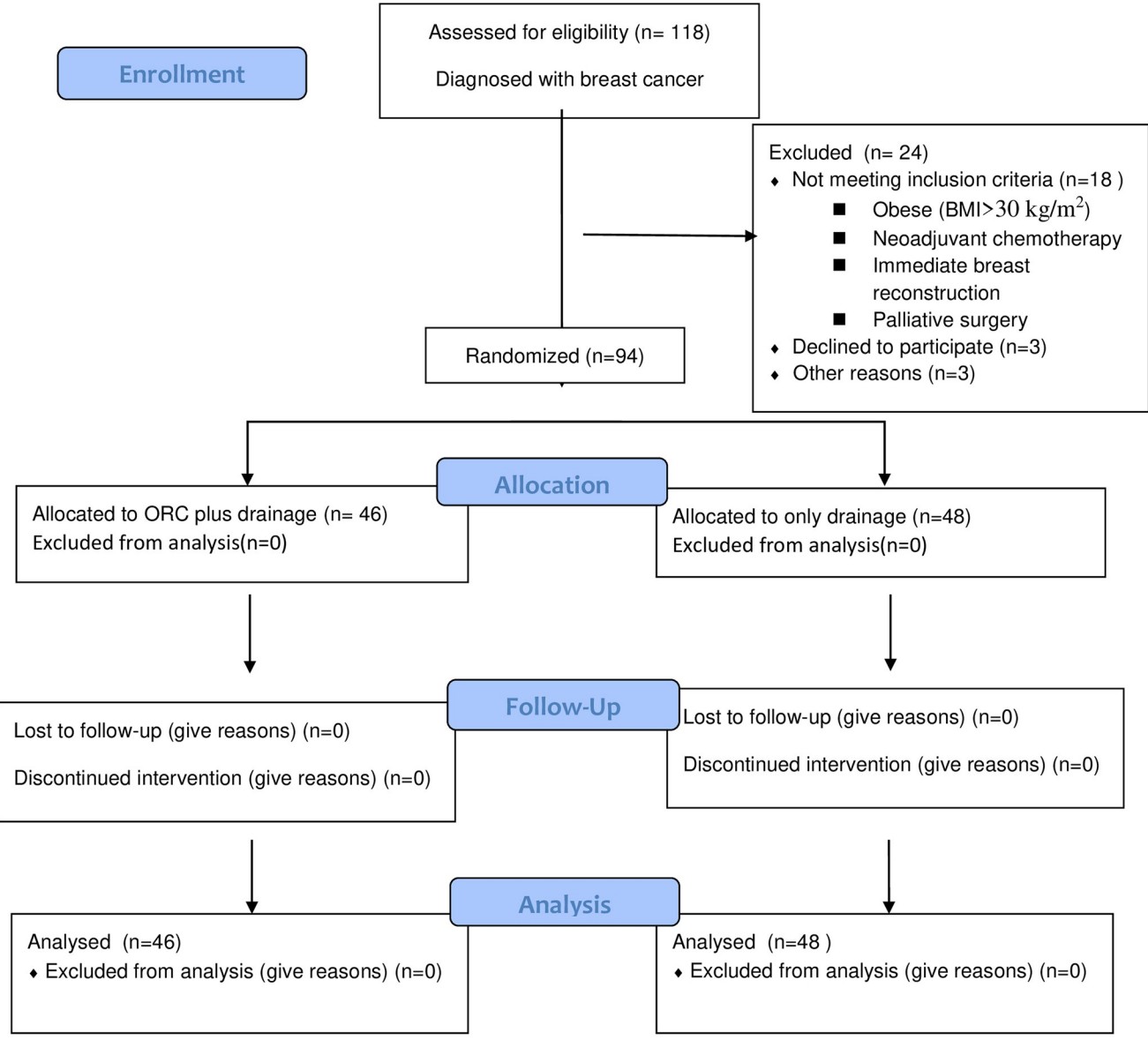

**Fig 1. CONSORT flow diagram for present study.**

are registered. The study protocol was approved by the Institutional Review Board of Gil hospital (IRB No. GCIRB 2019–150), and all patients provided written informed consent to a doctor, breast center office nurse or research nurse during preoperative clinic visits.

We recruited the following participants who were eligible for this trial. Patients were excluded if they had a personal history of hypersensitivity or allergic reaction to anticoagulants, were obese (defined as a body mass index $>30$ kg/m$^2$), had received neoadjuvant chemotherapy, or planned to undergo immediate breast reconstruction or palliative surgery. Obesity, surgery after chemotherapy, and palliative surgery are well-known risk factors for wound infection and larger seroma formation. Immediate breast reconstruction surgery is also possible risk of wound infection and seroma formation. Flow diagram of patients was shown in Fig 1.

## Interventions

All surgical procedures involved the same type of incision, using principally identical methods with total mastectomy and lymph node dissection performed at a standard using electrocautery and ultrasonic dissection technology that were used for hemostasis and lymphostasis [10]. Wounds were irrigated with 2L of normal saline prior to wound closure, with excess liquid removed by drying with pads. Two separate suction drainage tubes (Hemovac® 400ml compact evacuator) were inserted, one into the breast and the other into axillary dead space. To assess the efficacy of ORC, we defined the cases and controls as follows:

Case: Before wound closure, we splitted a 10x10 (cm) sized ORC gauze in half and implanted each on the mammary fold and axillary dead space, respectively. ORC may be left in a surgical site.

Control: Before wound closure, we did not implant ORC, letting only suction drainage tubes inserted.

## Outcomes (primary and secondary)

The drainage volume was measured daily at the same time during hospitalization. The drainage tubes were removed when the amount of drainage was below 30 ml/day on at least two consecutive days. Compressive bandages were maintained by all patients until hospital discharge. Each patient was examined in outpatient clinic after 7 days of discharge. If remnant fluid collection over the wound was identified, wound aspiration with disposable syringe was done and measured with reports. Total amount of drainage includes the amount of aspiration in the outpatient clinic as well. For analysis of ORC effect related to seroma formation, mean total drainage volume (ml) and duration of hospitalization (day) were recorded and compared with each group. Duration of hospitalization was coded as binary, based on POD 10 at study end.

## Sample size

G*Power software (version 3.1.9.2) was used to determine the number of patients needed per group. A priori power calculations estimated that a minimum of 45 subjects in each arm would enable us to detect the difference with 80% power (alpha = 0.05) with a standard deviation of approximately 15%. We presumed drop rate would be around 10%.

## Randomization (random number generation, allocation concealment, implementation)

Randomization in allocation to the 2 groups was achieved via a computer-generated program, which was the one treated with ORC plus closed suction drainage and the other with closed suction drainage alone. The randomization scheme utilized an allocation algorithm to ensure similar sample sizes at the end of patient accrual. Patients were randomized upon entering the operating room, at which time the surgeon opened the sealed envelope and read the group assignment card.

## Blinding

Participants(patients), practitioners, data collectors, outcome adjudicators, and researchers who analyze data were all blinded to their allocation throughout the course of the study.

### Statistical analysis

Categorical variables were compared by chi-square tests and continuous variables by Student's t tests. All statistical analyses were performed IBM SPSS Statistics 19 software, with a *P* value <0.05 considered statistically significant.

## Results

### Patient characteristics

A total of 94 patients were enrolled, including 60 (63.8%) who underwent total mastectomy with ALND and 34 (36.2%) who underwent total mastectomy with SLNB (Table 1). Following randomization, 46 patients, of mean age 59.29 ± 10.78 years, were treated with ORC plus drainage and 48, of mean age 58.18 ± 12.06 years, were treated with drainage alone. Mean tumor sizes in these two groups were 3.99 ± 3.67 cm and 3.56 ± 2.44 cm, respectively. Immunohistochemical tumor type was similar in the two groups, as were changes in hemoglobin

**Table 1. Baseline demographic and clinical characteristics of included patients.**

|  | ORC + Drainage Group (n = 46) | Drainage Alone Group (n = 48) | *P* value |
|---|---|---|---|
| Age, yr | 59.29 ± 10.78 | 58.18 ± 12.06 | 0.697 |
| BMI, kg/m$^2$ | 23.61 ± 3.53 | 24.11 ± 3.39 | 0.566 |
| Side |  |  | 0.300 |
| Right | 26 (23.5) | 22 (24.5) |  |
| Left | 20 (22.5) | 26 (23.5) |  |
| Type of surgery |  |  | 0.877 |
| Total mastectomy with ALND | 29 (29.4) | 31 (30.6) |  |
| Total mastectomy with SLNB | 17 (16.6) | 17 (17.4) |  |
| Tumor size (cm) | 3.99 ± 3.67 | 3.56 ± 2.44 | 0.575 |
| T stage |  |  | 0.864 |
| T1-2 | 38 (37.7) | 39 (39.3) |  |
| T3-4 | 8 (8.3) | 9 (8.7) |  |
| N stage |  |  | 0.720 |
| N0-1 | 32(32.8) | 35 (34.2) |  |
| N2-3 | 14 (13.2) | 13 (13.8) |  |
| Histologic type |  |  | 0.931 |
| IDC | 39 (39.2) | 41 (40.9) |  |
| Other | 7 (6.9) | 7 (7.2) |  |
| Immunohistochemical type |  |  | 0.751 |
| Luminal A | 12 (13.7) | 16 (14.3) |  |
| Luminal B | 16 (13.7) | 12 (14.3) |  |
| HR(-),HER-2 enriched | 10 (10.3) | 11 (10.7) |  |
| Triple negative | 8 (8.3) | 9 (8.7) |  |
| ASA class ($\geq$ 2) | 32 (34.0) | 36 (38.3) | 0.601 |
| Diabetes mellitus | 6 (6.4) | 5 (5.3) | 0.786 |
| Change of hemoglobin level (g/dl) |  |  |  |
| between Pre-op and POD 1 | 0.82 ± 0.55 | 0.93 ± 0.85 | 0.524 |
| between POD 1 and POD 2 | 1.68 ± 0.99 | 1.36 ± 0.88 | 0.167 |

Data are shown as mean ± standard deviation, or number (percent).

BMI, body mass index; SLNB, sentinel lymph node biopsy; IDC, intraductal carcinoma; DCIS, ductal carcinoma in situ; HR, hormone receptor; HER-2, human epidermal growth factor receptor 2; ASA, American Society of Anesthesiologists; POD, postoperative day.

**Table 2. Comparison of postoperative outcomes in two groups.**

|  | ORC + Drainage Group (n = 46) | Drainage Alone Group (n = 48) | *P* value |
|---|---|---|---|
| Drainage volume (ml) |  |  |  |
| Total volume | 1134 ± 507 | 1033 ± 643 | 0.486 |
| Until POD 1 | 123 ± 54 | 143 ± 104 | 0.345 |
| Until POD 2 | 326 ± 110 | 310 ± 140 | 0.602 |
| Until POD 3 | 493 ± 155 | 441 ± 192 | 0.238 |
| Until POD 7 | 883 ± 310 | 790 ± 378 | 0.283 |
| Until POD 14 | 1109 ± 472 | 997 ± 577 | 0.401 |
| Total aspirate volume (ml) | 156.6 ± 184.7 | 156.3 ± 277.7 | 0.995 |
| Number of aspirations | 3.61 ± 3.03 | 4.88 ± 5.7 | 0.273 |
| Duration of hospitalization (day) | 12.13 ± 3.69 | 11.94 ± 3.16 | 0.826 |
| Time to removal of drainage (day) | 12.39 ± 4.17 | 11.91 ± 3.13 | 0.603 |
| Wound complications | 4 (4.3) | 3 (3.2) | 0.500 |

Data are shown as mean ± standard deviation or number (percent).

POD, postoperative day.

concentration from the day prior to surgery to postoperative day (POD) 1 and from POD 1 to POD 2.

## Postoperative outcomes in the two groups

A comparison of postoperative outcomes in the two groups showed that mean total drainage volume was similar in patients treated with ORC plus drainage and those treated with drainage alone (1134 ± 507 ml vs 1033 ± 643 ml, *P* = 0.486; Table 2). The drainage volume was slightly lower in the former group on POD 1 (123 ± 54 vs 143 ± 104 ml), but was slightly higher at all other time points. Duration of hospitalization was also similar in these two groups (12.13 ± 3.69 days vs 11.94 ± 3.16 days, *P* = 0.826), as was total aspirated volume after discharge.

Comparing those two outcomes for duration of hospitalization based on POD 10 (within POD10 versus after POD10), the risk of seroma formation was reduced by 12% (95% CI 26% to 14%) in ORC plus drainage group. The absolute difference was -6.2%. But there showed no statistically significant difference between ORC+drainage and drainage alone groups (Table 3).

Seven patients, all without diabetes mellitus, experienced postoperative wound complications (infection, dehiscence and necrosis), four treated with ORC plus drainage and three treated with drainage alone. One patient who was treated with ORC plus drainage underwent wound revision due to skin necrosis, whereas the other six patients improved without surgery.

## Discussion

To our knowledge, this is the first clinical study evaluating the efficacy of SurgiGuard® in preventing seroma formation in breast cancer patients who underwent total mastectomy. We had

**Table 3. Both absolute and relative effect sizes between the two groups in patients who discharged within POD10.**

| Endpoint | Number(%) | | |
|---|---|---|---|
|  | ORC + Drainage Group (n = 46) | Drainage Alone Group (n = 48) | Risk difference (95% CI) |
| **Duration of hospitalization (within 10 days)** | 24(52.2) | 28(58.3) | 6.2(14–26) |

hypothesized that the hemostatic properties of ORC would prevent postoperative accumulation of exudate and reduce capillary and lymphatic drainage, thereby reducing the risk of seroma formation. The results of this trial, however, failed to show that ORC significantly decreased wound drainage.

Other ORC-related studies showed that ORC has no beneficial effect of seroma reduction in their comparative studies. Chang et al. reported that there was no significant postoperative drainage reduction in the hepatic surgery [11]. Franceschini et al. demonstrated significant seroma formation in 45% of ORC placement group in their 2012 report. They discussed this seroma appeared in the early postoperative period as consequence of redundant ORC digestion, normally resolved within few weeks with repeated percutaneous aspiration [12].

The ability of other topical agents to reduce seroma formation has also been evaluated. For example, several studies have evaluated the use of fibrin glue, with some reporting that fibrin glue had no effect [13,14] and others reporting that fibrin glue had positive effects in reducing drainage [15,16]. Thrombin treatment also failed to show efficacy in reducing seroma formation [17,18]. However, differences among study methods, sample sizes and patient populations have limited the ability to draw meaningful conclusions. To date, however, no single agent has been found to be optimal in preventing seroma formation.

In our study, patients in both groups developed seromas. Furthermore, mean total drainage volume and mean number of aspirations after discharge were similar in the two groups. The use of ORC only slightly reduced the incidence of postoperative seroma formation. Calculation with effect size was done for duration of hospitalization based on POD 10, the risk of seroma formation was not reduced in the group of ORC use. This clinically unimportant reduction was not sufficient to justify the routine use of ORC.

Studies have shown that the use of SurgiGuard® after surgery could achieve hemostasis in pigs [19], and that ORC could prevent seroma formation or reduce the volume of postoperative drainage following submandibular gland surgery [20]. ORC has been reported to act as a reconstructive biomaterial to optimize esthetic results after oncoplastic procedures in breast cancer surgery [21–23]. By minimizing possible postoperative complications after mammaplasties, ORC may improve cosmetic outcomes. Previous trials, however, have been small and under-powered, and lacked the methodological rigor required to draw firm conclusions.

The clinicopathologic profiles of the patients in the two groups did not differ significantly. Even though 33 of the 94 patients had the N2 or HER2 positive or triple negative subtype, they were scheduled not to receive neoadjuvant chemotherapy and therefore enrolled in the study. In a meta-analysis, the rate of lymphedema was found to be higher among patients with advanced-stage tumors who had undergone more aggressive axillary node dissection and neoadjuvant chemotherapy [24]. We excluded patients who received neoadjuvant chemotherapy to control for selective bias. We demonstrated that tumor stage and tumor subtype were not related to seroma formation.

Mean total seroma volume in the present study was about three times higher than volumes reported in other studies [14,25], and median duration of hospitalization in the present study was about 12 days, which is longer than in other studies [6,14]. Daily seroma volume measurement is critical and should be precise for this study, participants was given for informed consent for longer hospitalization and examined thoroughly under daily inspection. The higher mean total seroma volume observed in our patients may have been due to longer duration of hospitalization with drainage tubes, and/or to a higher proportion of advanced-stage patients in the study population.

Longer drainage time is an well-known risk factor for wound infection [26,27], surgical site infection (SSI) following breast cancer surgery is not common albeit secondary to their rare occurrence. A participant who received wound revision due to skin necrosis had a history of

heart failure with medication. But, other patients with SSI had good performance before and after surgery. Additional studies in larger cohorts would be needed to draw definitive conclusions. The longer duration of hospitalization in the present work may have arisen because patients were not discharged until they were free from wound problems, such as infection and skin necrosis, and drainage tubes had been removed. Keeping patients until wound complications had resolved added three days to the duration of hospitalization on average. The participant with skin necrosis in ORC group was treated with revision surgery and postoperative oral antibiotics during 3 days. There was no additional revision procedure and showed quick recovery after treatment. Participants who experienced SSI experienced no larger amount of seroma in both groups. In addition, we found that the rate of wound complications did not differ between the groups that did and did not use ORC.

ORC has a possible role as a reconstructive biomaterial and a few reports in the plastic surgery literature present the improvement of aesthetic outcomes in patients undergoing oncoplastic procedures for breast cancer. ORC integration and digestion processes are observed by diffuse fibrosis and homogeneous neovascularization within the construct [12,28,29]. ORC may be used to help manage exudate and promote granulation tissue development and moist wound healing [30]. Despite this advantageous feature of ORC, some studies report a foreign body reaction in various surgical site due to incomplete bioabsorption [31–33].

Although we found that SurgiGuard® did not significantly alter the risk of seroma formation, the limit of our study is that the outcomes depend only on Korean populations. Larger clinical trials are required to fully evaluate the hemostatic effects of ORC on exudate accumulation and wound healing, as well as their potential correlation with seroma formation, in patients undergoing mastectomy with axillary operations.

## Supporting information

**S1 Checklist. CONSORT 2010 checklist.**
(DOC)

**S1 File. Research proposal_Eng.**
(DOCX)

**S2 File. Research proposal_Korean.**
(DOCX)

## Author Contributions

**Conceptualization:** Yun Yeong Kim.

**Data curation:** Kug Hyun Nam, Joon-Hyop Lee.

**Formal analysis:** Joon-Hyop Lee.

**Funding acquisition:** Yun Yeong Kim.

**Methodology:** Kug Hyun Nam, Yoo Seung Chung.

**Project administration:** Yoo Seung Chung.

**Resources:** Yong Soon Chun, Heung Kyu Park.

**Supervision:** Heung Kyu Park, Yun Yeong Kim.

**Validation:** Yong Soon Chun.

**Visualization:** Kug Hyun Nam.

**Writing – original draft:** Kug Hyun Nam.

**Writing – review & editing:** Yun Yeong Kim.

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
