## [Decision Letter · Decision Letter 0]

13 Apr 2021

PONE-D-20-33439

The efficacy of oxidized regenerated cellulose (SurgiGuard®) in breast cancer patients who undergo total mastectomy with node surgery: A prospective randomized study in 94 patients

PLOS ONE

Dear Dr. Kim,

Thank you for submitting your manuscript to PLOS ONE. After careful consideration, we feel that it has merit but does not fully meet PLOS ONE’s publication criteria as it currently stands. Therefore, we invite you to submit a revised version of the manuscript that addresses the points raised during the review process.

The manuscript has been evaluated by three reviewers, and their comments are available below.

The reviewers have raised a number of concerns that need attention. They request additional information on methodological aspects of the study and statistical analyses. They also request revisions to the conclusions to ensure that they are presented appropriately.

Could you please revise the manuscript to carefully address the concerns raised?

We note that one or more reviewers has recommended that you cite specific previously published works. As always, we recommend that you please review and evaluate the requested works to determine whether they are relevant and should be cited. It is not a requirement to cite these works. We appreciate your attention to this request.

We look forward to receiving your revised manuscript.

Kind regards,

Marianne Clemence

Associate Editor

PLOS ONE

Journal Requirements:

2. Thank you for submitting your clinical trial to PLOS ONE and for providing the name of the registry and the registration number. The information in the registry entry suggests that your trial was registered after patient recruitment began. PLOS ONE strongly encourages authors to register all trials before recruiting the first participant in a study.

i) your reasons for your delay in registering this study (after enrolment of participants started);

ii) confirmation that all related trials are registered by stating: “The authors confirm that all ongoing and related trials for this drug/intervention are registered”.

3. In your Methods section, please provide additional information about the participant recruitment method and the demographic details of your participants. Please ensure you have provided sufficient details to replicate the analyses such as a description of how participants were recruited.

4. Please ensure you have discussed any potential limitations of your study in the Discussion, including study design, sample size and/or potential confounders.

5. In the Methods section, please state your primary and any secondary outcomes.

7. Please include a separate caption for each figure in your manuscript.

8. Please include captions for your Supporting Information files at the end of your manuscript, and update any in-text citations to match accordingly. Please see our Supporting Information guidelines for more information: http://journals.plos.org/plosone/s/supporting-information

Reviewers' comments:

Reviewer's Responses to Questions

**Comments to the Author**

1. Is the manuscript technically sound, and do the data support the conclusions?

Reviewer #1: Yes

Reviewer #2: Partly

Reviewer #3: Partly

2. Has the statistical analysis been performed appropriately and rigorously? 

Reviewer #1: Yes

Reviewer #2: No

Reviewer #3: Yes

3. Have the authors made all data underlying the findings in their manuscript fully available?

Reviewer #1: Yes

Reviewer #2: No

Reviewer #3: Yes

4. Is the manuscript presented in an intelligible fashion and written in standard English?

Reviewer #1: Yes

Reviewer #2: Yes

Reviewer #3: Yes

5. Review Comments to the Author

Reviewer #1: Manuscript Number: PONE-D-20-33439

Manuscript Title: The efficacy of oxidized regenerated cellulose (SurgiGuard®) in breast cancer patients who undergo total mastectomy with node surgery: A prospective randomized study in 94 patients

This study is a single blinded, prospective randomized controlled trial conducted at a single specialized breast cancer center in order to value the efficacy of oxidized regenerated cellulose (ORC) in breast cancer patients who undergo total mastectomy with node surgery.

This study was registered at Clinical Research Information Service, Republic of Korea (KCT0005637).

Ninety four breast cancer patients were enrolled in this study. The patients were randomized into two groups, one treated with ORC plus closed suction drainage and the other with closed suction drainage alone.

After an analysis of the results, the Authors conclude that use of ORC did not significantly alter the risk of seroma formation.

The manuscript falls under the scope of journal; It is an interesting topic. The study presents the results of original research.

The article is presented in an intelligible fashion; The work is described expansively; it is well structured and developed.

The manuscript is written in standard English; Language and grammar are acceptable. There are some typos.

The abstract is appropriate with the study; The introduction is adequate and clear. Materials and Methods are clear and linear.

Experiments, statistics, and analyses are performed to a technical standard and are described in sufficient detail; Categorical variables were compared by chi-square tests and continuous variables by Student’s t tests. All statistical analyses were performed IBM SPSS statistics 19 software, with a p value <0.05 considered statistically significant.

Results reported have not been published elsewhere.

The discussion and conclusions should be improved:

- Discussion: the Authors should describe better the benefits and issues due to the use of ORC in breast surgery (see new references); description of histologic features resulting from the interaction between tissues and ORC may be very important in understanding the physiologic processes behind ORC integration and local side effects.

- The conclusions are supported by the data but should be better explained and presented in an appropriate fashion.

Some specific references about use of ORC in breast surgery should be read and added in the discussion:

- Franceschini G, Visconti G, Sanchez AM, Di Leone A, Salgarello M, Masetti R. Oxidized regenerated cellulose in breast surgery: experimental model. J Surg Res. 2015 Sep;198(1):237-44. doi: 10.1016/j.jss.2015.05.012.

- Franceschini G. Internal surgical use of biodegradable carbohydrate polymers. Warning for a conscious and proper use of oxidized regenerated cellulose. Carbohydr Polym. 2019 Jul 15;216:213-216. doi: 10.1016/j.carbpol.2019.04.036.

- Rassu PC. Observed outcomes on the use of oxidized and regenerated cellulose polymer for breast conserving surgery - A case series. Ann Med Surg (Lond). 2015 Dec 22;5:57-66. doi: 10.1016/j.amsu.2015.12.050.

- Rassu PC, Serventi A, Giaminardi E, Ferrero I, Tava P. Use of oxidized and regenerated cellulose polymer in oncoplastic breast surgery. Ann Ital Chir. 2013 Jan 29;84(ePub):S2239253X13020288.

- Franceschini G, Visconti G, Terribile D, Fabbri C, Magno S, Di Leone A, Salgarello M, Masetti R. The role of oxidized regenerate cellulose to prevent cosmetic defects in oncoplastic breast surgery. Eur Rev Med Pharmacol Sci. 2012 Jul;16(7):966-71.

- Franceschini G, Sanchez AM, Visconti G, Di Leone A, Salgarello M, Masetti R. Quadrantectomy with oxidized regenerated cellulose ("QUORC"): an innovative oncoplastic technique in breast conserving surgery. Ann Ital Chir. 2015;86:548-52.

CONSORT checklist:

Outcomes (Item 6a): The Authors should better describe outcomes and completely defined pre-specified primary outcome measure including how and when it was assessed.

Sample size (Item 7a): it was determined by

G*Power software (version 3.1.9.2). A priori power calculations estimated that a minimum of 45 subjects in each arm would enable to detect the difference with 80% power (alpha= 0.05) with a standard deviation of approximately 15%. The Authors presumed drop rate would be around 10%.

Sequence generation (Item 8a), Allocation concealment (Item 9), Blinding (Item 11a): The randomization scheme utilized an allocation algorithm to ensure similar sample sizes at the end of patient accrual. Patients were randomized upon entering the operating room, at which time the surgeon opened the sealed envelope and read the group assignment card. Patients were blinded to their allocation throughout the course of the study.

However, the type of allocation algorithm should be better described; are outcome assessors blinded?

Outcomes and estimation (Item 17a/b): they should be revised.

Harms (Items 19): All important harms and unintended effects in each group are reported.

Registration (Item 23): This study was registered at Clinical Research Information Service (CRiS), Republic of Korea (KCT0005637).

Protocol (Item 24): The study protocol was approved by the Institutional Review Board of Gil hospital (IRB No. GCIRB 2019-150).

Funding (Item 25): This work was supported by Gachon University research fund of 2017 (GCU-2017- 5258). However, the role of funders

should be better described.

The manuscript may be accepted with some revisions.

Reviewer #2: The objective of this prospective, randomized controlled trial (RCT) is to assess the effectiveness of the "ORC + closed suction drainage", over the "only closed suction drainage" procedures in breast cancer patients who underwent total mastectomy. The study was registered as a RCT within the Korean Clinical Trial Registry (with a legit number), and was approved by the respective IRB/Ethics Committee. While the study objectives sound interesting, is important, and on target, some shortcomings were observed, in regards to abiding by the CONSORT guidelines for conducting and reporting results of high-quality randomized controlled trials (RCTs). Some other (statistical) comments were also provided.

1. Methods:

Methods reporting appeared very messy. An orderly manner is suggested, following CONSORT guidelines, without repeating information, such as Trial Design, Participant Eligibility Crtieria and settings, Interventions, Outcomes, sample size/power considerations, Interim analysis and stopping rules, Randomization (details on random number generation, allocation concealment, implementation), Blinding issues, etc, should be mentioned. The authors are advised to create separate subsections for each of the possible topics (whichever necessary), and that way produce a very clear writeup. I see the Authors indeed made an attempt; a nice example can be seen in the manuscript below:

https://www.sciencedirect.com/science/article/pii/S0889540619300010

Specific comments below:

(a) For instance, the randomization and allocation concealment should be made very clear (they are NOT the same thing); the trial staff recruiting patients should NOT have the randomization list. Randomization should be prepared by the trial statistician, and he/she would not participate in the recruiting.

(b) More details on the randomization is necessary. For example, was it a block randomization?. If Yes, then what's the block size?

(c) Sample size/power: The sample size/power computations should be conducted using the primary response; it is not clear what the authors did. Furthermore, the calculations should clearly state the "effect size" under consideration. Taking a standard deviation of ~ 15% doesn't convey much information.

(d) Statistical Analysis: Overall, looks OK, and straightforward. Standard 2-sample t-tests were used. Did the authors confirm that Gaussian assumptions were valid? Otherwise, they would have resorted to nonparametric tests, such as various Wilcoxon tests.

2. Results & Conclusions:

(a) The authors should check that any statement of significance should be followed by a p-value in the entire Results section.

(b) Null findings were observed (no differences between the groups). The Conclusions section should clearly state that the results/conclusions are "only" from this Korean population, and allude to future studies, sometimes multicenter, with possibly higher sample sizes, and/or combining other populations to evaluate the difference further.

Reviewer #3: Dear Author,

This is a well written paper on a subject that is highly relevant for breast cancer patients undergoing mastectomy. Seroma formation results in the risk of SSI, more visits and interventions, higher costs, decreased satisfaction and potential delay in adjuvant treatment and/or return to work. Several interventions have been investigated over the years, but results are not very satisfying. Although the methods are well described there are some major flaws in the methods chosen and some important data/analysis is lacking. If the below comments are addressed well I would think this paper is relevant and should definitely be published. Especially because wound care intervention studies are scarce, especially RCTs. It is important to also publish negative studies, as future studies could repeat this intervention with improved methods potentially finding a benefit of this theoretical smart and patient friendly intervention.

Introduction

Please add a paragraph on the relevance of seroma formation for patients and healthcare. More and longer seroma duration has the risk of SSI, more visits and interventions, higher costs, decreased satisfaction and potential delay in adjuvant treatment and/or reteurn to work.

Methods

1. The intervention itself is not clearly described. In the current manuscript just very briefly the SurgiGuard product working mechanism is described in the introduction, but the exact application is unclear for a reader. please ellaborate more on this, as this is crucial for the potential user group. I assume the product is left inthe wound bed before closing the mastectomy wound. Please also describe in detail the surgical procedure. Wat settings of electrocautery were used? What kind of suture technique?

2. exclusion criteria: why is a BMI >30 used as an exclusion. It is a known risk factor for wound infection and seroma formation.

3. Wound complications: this subject should be discussed in more detail. What specific wound complications: infection (according to which criteria, preferrably CDC criteria), dehiscense, necrosis. This is very relevant as infections are the major cause of morbidity and costs in this patient group. Also the occurence of seroma/hematoma is highly correlated to the occurence of wound infection. (refs El-Tamer 2007, Xue 2012, Mukesh 2012 and Struik 2018)

4. I would be interested in postoperative Hemoglobine decline and did you collect data on blood loss during surgery and compare this between groups?

Results

1. There is a strikingly long mean drainage time in your study, compared to other studies. Long drainage time is a known risk factor for infection and should therefore be avoided. Also this results in very long mean hospital stays. Please ellaborate on this more and compare to literature. The long drainage time is probably impacting on your results. DId you consider to use other endpoints such as CDC criteria for seroma, number of reintervention and f.e. US measurements of seroma volumes instead?

Discussion

please compare especially drainage time, SSI and potential use of other seroma related outcomes to literature.

6. PLOS authors have the option to publish the peer review history of their article (what does this mean?). If published, this will include your full peer review and any attached files.

Reviewer #1: No

Reviewer #2: No

Reviewer #3: No

---

## [Author Response · Author response to Decision Letter 0]

31 May 2021

I checked all the review comments and responded point by point and uploaded response file. Thank you.

---

## [Decision Letter · Decision Letter 1]

10 Jan 2022

PONE-D-20-33439R1

The efficacy of oxidized regenerated cellulose (SurgiGuard®) in breast cancer patients who undergo total mastectomy with node surgery: A prospective randomized study in 94 patients

PLOS ONE

Dear Dr. Kim,

Thank you for submitting your manuscript to PLOS ONE. After careful consideration, we feel that it has merit but does not fully meet PLOS ONE’s publication criteria as it currently stands. Therefore, we invite you to submit a revised version of the manuscript that addresses the points raised during the review process.

Thank you for your patience during the peer review process of your submission. As you may see some previous reviewers for the manuscript were unavailable to comment on the revised version of the submission. As such we have sought the opinions of a 4^th^ reviewer.

The reviewer had raised some concerns which require attention, their comments may be seen below.   In particular the reviewer feels that additional details on the theoretical frameworks used to develop the study rational as well as the study design is required.

Could you please revise the manuscript to carefully address the concerns raised?

We look forward to receiving your revised manuscript.

Kind regards,

Lucinda Shen, MSc

Staff Editor

PLOS ONE

Reviewers' comments:

Reviewer's Responses to Questions

**Comments to the Author**

1. If the authors have adequately addressed your comments raised in a previous round of review and you feel that this manuscript is now acceptable for publication, you may indicate that here to bypass the “Comments to the Author” section, enter your conflict of interest statement in the “Confidential to Editor” section, and submit your "Accept" recommendation.

Reviewer #2: All comments have been addressed

Reviewer #4: (No Response)

2. Is the manuscript technically sound, and do the data support the conclusions?

Reviewer #2: (No Response)

Reviewer #4: Partly

3. Has the statistical analysis been performed appropriately and rigorously? 

Reviewer #2: (No Response)

Reviewer #4: Yes

4. Have the authors made all data underlying the findings in their manuscript fully available?

Reviewer #2: (No Response)

Reviewer #4: No

5. Is the manuscript presented in an intelligible fashion and written in standard English?

Reviewer #2: (No Response)

Reviewer #4: Yes

6. Review Comments to the Author

Reviewer #2: (No Response)

Reviewer #4: COMMENTS TO THE AUTHORS

This is a very good initiative and it will be an initial step in determining the role of ORC in seroma prevention after mastectomy. Studies like this will go a long way in reducing the ever-present problem of seroma after mastectomy but there some clarifications we need from the author before publication

1. The authors registered the clinical trial on 25th November, 2020 while the patient enrollment started from 01st June 2019 to 31st August 2020. The registration was done 4 months after completion of the study which is against the ICMJE’s clinical trial registration policy. There is need for the clarification from the authors.

2. It is evident that the authors have a good grasp of seroma, but as part of the introduction, there is need to briefly mention proposed theories for development of seroma after mastectomy. Also, there is need to briefly mention the effect of seroma on patients, surgical team and the hospital community.

3. The authors mention high drain output and prolonged drainage as risk for seroma development after mastectomy. However, there is an oversight as the reference for this wasn’t cited.

4. The authors description of data collection protocol should be detailed and should include detailed preoperative data collection protocol, detailed intraoperative protocol including the protocol for placement of ORC in the patients, and detailed post-operative protocol including the protocol for diagnosing seroma.

5. The authors also need to be detailed in the description of the study design. Is it a superiority design or a non-inferiority design?

6. There is need for the authors to highlight the incidence of seroma in each group and the overall incidence of seroma in the study.

7. Wound complications developed by some patients should be specified and the measures taken to handle them should be highlighted.

8. The authors gave a detailed discussion but there is need to organize it better to improve flow and readability. Also, there is no need to discuss lymphedema as part of the manuscript.

7. PLOS authors have the option to publish the peer review history of their article (what does this mean?). If published, this will include your full peer review and any attached files.

Reviewer #2: No

Reviewer #4: No

---

## [Author Response · Author response to Decision Letter 1]

15 Feb 2022

We thank the reviewers and editors for their efforts and essential comments. Our replies follow in the order of each reviewer's point. We attach 'the response to reviewer' file to the system.

---

## [Editor Report · Decision Letter 2]

28 Feb 2022

PONE-D-20-33439R2The efficacy of oxidized regenerated cellulose (SurgiGuard®) in breast cancer patients who undergo total mastectomy with node surgery: A prospective randomized study in 94 patientsPLOS ONE

Dear Dr. Kim,

Thank you for submitting your manuscript to PLOS ONE. After careful consideration, we feel that it has merit but does not fully meet PLOS ONE’s publication criteria as it currently stands. Therefore, we invite you to submit a revised version of the manuscript that addresses the points raised during the review process.

As part of the intervention, you need to mention how you perform the mastectomy by citing an article that describe the procedure and making a statement for the readers to refer to the said article.

In discussion, compare your findings to other studies from Africa, Europe, and Americas

We look forward to receiving your revised manuscript.

Kind regards,

IBRAHIM UMAR GARZALI, MBBS

Academic Editor

PLOS ONE

Journal Requirements:

Additional Editor Comments (if provided):

Dear authors,

You have addressed most of my concerns as requested. However, in the intervention, you may need to mention how the mastectomy was performed. You don't need to describe the procedure, just refer your readers to a manuscript that describe the procedure and cite it in you article. Also in your discussion, it will be good if you compare your findings to other studies in Africa, Europe and Americas.

Thank you.
---

## [Author Response · Author response to Decision Letter 2]

9 Apr 2022

Thank you for your kind comments. According to your recommendations, we added a mention about mastectomy in reference to conventional procedure website. Plus, we included other two similar 

comparison studies documenting especially about ‘seroma formation’, those are in few of ORC-related 

randomized controlled studies until now. 

Finally, we reviewed reference list to ensure that it is compete and correct. 

Thank you.

---

## [Editor Report · Decision Letter 3]

14 Apr 2022

The efficacy of oxidized regenerated cellulose (SurgiGuard®) in breast cancer patients who undergo total mastectomy with node surgery: A prospective randomized study in 94 patients

PONE-D-20-33439R3

Dear Dr. Kim,

We’re pleased to inform you that your manuscript has been judged scientifically suitable for publication and will be formally accepted for publication once it meets all outstanding technical requirements.

Kind regards,

IBRAHIM UMAR GARZALI, MBBS, FWACS

Guest Editor

PLOS ONE
---

## [Editor Report · Acceptance letter]

18 May 2022

PONE-D-20-33439R3 

The efficacy of oxidized regenerated cellulose (SurgiGuard) in breast cancer patients who undergo total mastectomy with node surgery: A prospective randomized study in 94 patients 

Dear Dr. Kim:

I'm pleased to inform you that your manuscript has been deemed suitable for publication in PLOS ONE. Congratulations! Your manuscript is now with our production department. 

Kind regards, 

on behalf of

Dr. IBRAHIM UMAR GARZALI 

Guest Editor

PLOS ONE